# The association between nurses' physical activity counselling and patients' perceptions of care quality in a primary care facility in Ghana

Nestor Asiamah[1,2]*, Emmanuel Opoku[3], Kyriakos Kouveliotis[4]

1 Division of Interdisciplinary Research and Practice, School of Health and Social Care, University of Essex, Colchester, United Kingdom, 2 Department of Health Services Research, Africa Centre for Epidemiology, Accra, Ghana, 3 Department of Marketing, Accra Technical University, Accra Metro, Accra, Ghana, 4 Department of Health Care Management, International Telematic University Uninettuno, Italy, Rome

* n.asiamah@essex.ac.uk, nestor.asiamah@ace-gh.org

**Data Availability Statement:** Data are attached as supplemental information (SPSS file).

**Funding:** The authors received no specific funding for this work.

## Abstract

Many countries including Ghana and Australia have adopted physical activity (PA) counselling in healthcare as a public health improvement strategy. Even so, more evidence is needed to improve clinical PA counselling among clinicians, including nurses. This study examined the association between nurses' physical activity counselling (NPAC) and patients' perceptions of care quality. The study adopted a cross-sectional design with a sensitivity analysis against potential confounding. The setting of the study was a public primary care facility in Darkuman, Accra. Participants were 605 patients in wards and the Outpatient Department of the facility. Data were collected using a self-reported questionnaire and analyzed using structural equation modeling. A sensitivity analysis was conducted to select potential confounding variables for the study. The study found that higher care quality was associated with larger scores of NPAC (β = 0.34; CR = 8.65; p = 0.000). NPAC has no significant direct association with patient satisfaction (β = 0.01; CR = 0.22; p > 0.05) and loyalty (β = 0.05; CR = 1.21; p > 0.05), but care quality and patient satisfaction fully mediate the association between NPAC and patient loyalty. It is concluded that NPAC in healthcare can improve care quality and indirectly increase patient satisfaction and loyalty through care quality. The incorporation of PA counselling into clinical nursing may, therefore, be consistent with the core mission of hospitals.

## Introduction

The health benefits of physical activity (PA) have been reported by many researchers in different disciplines [1–4]. Specifically, PA has been evidenced to protect against cardiovascular diseases [5, 6], neurodegenerative diseases [1, 3, 4] and other non-infectious diseases [7, 8] PA has also been found in research to reduce the risk of mortality in the general population [9, 10] and benefit healthy aging [11]. The promotion of PA as a healthy behavior in all segments of

**Competing interests:** The authors have no competing interests

**Abbreviations:** NP1-NP10, Items on NPAC; PL1-PL3, Items on patient loyalty; PS, Patient satisfaction; PL, Patient loyalty; SQ1-SQ26, Items on care quality; SQ, Service quality (Care quality).

the population is, therefore, a top agenda in public health. Many PA promotion programs and interventions have been implemented at the regional and national levels, with a popular example being the World Health Organization's PA promotion framework for the European region [12]. Some of these interventions are the inclusion of patients' PA level into medical records, creation of sports and exercise laboratories in hospitals where they are not available, and the promotion of PA as a healthy behavior in healthcare [13]. The World Health Organization's PA promotion framework for the European region has benefitted clinical PA promotion for a couple of reasons. Firstly, much of research investigating whether PA promotion (e.g., PA counselling) in healthcare is important and beneficial to patients has been funded and championed by the WHO through the foregoing framework [14], which means that the program has contributed to evidence needed to implement PA counselling in healthcare. Secondly, the adoption of PA counselling in clinical practice in many European countries has been based on the framework's recommendation [12, 14]. Despite these and similar efforts, the global prevalence of insufficient PA is still high [7, 8, 14], with developed countries including Kuwait, Canada, and Iceland accounting for a PA insufficiency rate of more than 50% [14]. This trend has dire implications for global health, including an increased burden of disease and healthcare expenditure [13, 14].

Many countries including Ghana, Australia, and United Kingdom have rolled out interventions prescribing PA in primary care facilities [13]. In these programs, nurses discuss PA with patients, focusing on standard PA recommendations and the general health benefits of PA [15–17]. Some researchers have, however, opposed this program, contending that nurses are not well suited for PA counselling [7, 8, 13]. Others in agreement to this argument opined that the program has nothing or little to do with the mission of healthcare facilities and will shift the focus of healthcare [13, 18]. We are however of the view that PA counselling in healthcare can rather increase the value of healthcare because most patients consider PA a behavior that benefits personal health and, therefore, attach importance to any program guiding uptake of PA [7, 13, 17].

In this study, therefore, we examined the association between nurses' PA counselling and patients' perceptions of care quality in the context of a public health facility. Since care quality is the foundation of patients' satisfaction and continuous utilization of healthcare [19], we attempted to provide holistic evidence by assessing whether patient satisfaction and loyalty are outcomes of care quality linked to PA counselling. Moreover, patient loyalty is a measure of long-term service utilization [19, 20], so its inclusion in this study was a way to investigate whether PA counselling by nurses can be associated with long-term utilization of services by patients.

## Methods

### Design

This study adopted a cross-sectional design with sensitivity analyses against potential confounding.

### Population, sample, and selection

The study population was patients in the OPD (Outpatient Department) and wards of Oduman Polyclinic, a 30-bed primary healthcare facility. We focused on patients in the OPD because in-patients in the facility were frail and did not have the opportunity to exercise or practice PA-related advice from clinical nursing. Moreover, in-patients in the facility did not meet all our selection criteria, which are reported below. Participants' age ranged between 19 and 76 years.

We selected participants with the following inclusion criteria: (1) willingness to participate in the study; (2) being aged 18 years or higher; (3) being available to complete the survey during a defined period; (4) not having any physical and mental condition that precluded PA; (5) having a minimum of a basic education (e.g., basic school leaving certificate), which evidenced the ability to complete the survey in English; (6) having previously received health care, including PA counselling, in the facility for at least a year, and (7) being part of the registry of the facility as a regular patient. Physicians in the clinic followed medical standards to identify those who met this fourth criterion. The number of patients who satisfied all criteria was 621. We subsequently used the G*Power 3.1.9.4 software with relevant statistics (i.e., effect size = 0.4; σ = 5%; power = 0.8) from a related study [21] to calculate the minimum sample size required for the study. The minimum sample size reached for structural equation modelling (SEM) was 44. Because SEM produces the best results with a sample size ≥500 [22], we decided to gather data on all 621 patients recruited. In recruiting participants, we contacted each patient who was part of the clinic's registry via a phone call or email to discuss the purpose and benefits of the study as well as other research conditions and plans (e.g., data collection period). Those who agreed to participate were then screened based on the selection criteria.

## Measures and operationalization

NPAC was measured using a 7-item scale developed by Asiamah et al. [23] based on the study's sample. This instrument comprises two domains, namely *PA recommendation* (i.e. 4 items measuring how often nurses mentioned PA as a healthy and important habit for patients) and *follow-up* (i.e. how frequently the nurse followed up with patients to know about progress and developments in their PA). This domain constituted 3 items shown in S1 Appendix. The above measure, which has three descriptive anchors (i.e. *not at all*– 1; *sometimes*– 2; and *always*– 3), was used because it is the only available tool for measuring nurses' PA counselling in healthcare and has produced satisfactory psychometric properties, specifically validity (i.e. discriminant and convergent validity) and internal consistency assessed with Cronbach's α coefficient. In the current study, the scale produced a Cronbach's α value of 0.725 (follow-up = 0.721; and PA recommendation = 0.707), which satisfies the recommended criterion of α ≥0.7 [22]. In measuring PA counselling, we asked patients to indicate how often nurses in the chosen facility provided pieces of advice (measured by the foregoing scale) regarding PA. The specific pieces of advice provided by nurses are the 7 statements shown in S1 Appendix.

Patients were further asked to report their perceptions regarding the quality of services delivered at the health facility, taking into account their experience with PA counselling. Health care quality was measured using HEALTHQUAL, a 26-item scale developed by Lee [19] and comprised five domains (i.e., empathy, tangibles, safety, efficiency, and quality care improvement). The participants' response to items of the scale was based on five descriptive anchors [very bad (1), bad (2), somewhat good (3), good (4), and very good (5)]. HEALTHQUAL originally produced a Cronbach's alpha α ≥0.8 for each domain and α>0.9 for the whole scale [19]. In the current sample, it produced a Cronbach's α coefficient ≥0.7 for the domains and α = 0.889 for the whole scale. HEALTHQUAL was preferred to other available scales because it includes a domain measuring the degree of care quality improvement. Patient satisfaction and loyalty were measured using two and three items respectively borrowed from Sharma [20]. Items used to measure care quality, patient satisfaction, and patient loyalty are shown in S2 Appendix. The scale used to measure patient loyalty produced a Cronbach's α = 0.856 and factors loadings ≥ 0.5 whereas that of patient satisfaction yielded a Cronbach's α = 0.811 and factor loadings ≥0.5. Gender, education, NHIS (National Health Insurance Scheme) status, and age were measured as potential covariates. Gender (male vs. female) and NHIS

(NHIS subscriber vs. not NHIS subscriber) were measured as dichotomous variables and were dummy coded twice in data analysis. Education was measured as the highest formal education qualification acquired by the patient while individual income was measured as the individual's monthly take-home pay (₵).

### Data collection process

This study was approved by the management of the hospital and received ethical clearance from Africa Center for Epidemiology with Institutional Review Board number ACE-EPP-2019. There was no deviation in the study protocol after ethics approval was received. Questionnaires were administered through hand delivery at the hospital over a month (September 1 to October 3, 2019). Three (3) trained field workers assisted in administering questionnaires. All respondents participated in the study voluntarily after signing an informed consent form that detailed the purpose and benefits of the study. Participants were encouraged to complete questionnaires immediately to maximize response rate, but many of them took over two weeks to complete and return questionnaires through a courier hired by the researchers. Completed questionnaires were returned in stamped and sealed envelopes provided by the researchers.

### Statistical analysis approach

The study employed SPSS for Windows 25 (IBM SPSS Inc., New York, U.S.A.) and its Amos software to analyze the data. Data were analyzed in two phases. In the first exploratory phase, data were summarized using descriptive statistics (i.e., frequency, percent, mean, standard deviation, kurtosis, and skewness). Estimates of skewness and kurtosis were found to meet recommended levels [24, 25] and thus signified the absence of outliers in the data. Multivariate normality of the data, which is a requirement for structural equation modelling (SEM) [24, 26], was met with the criterion p2>0.05 [27], where p2 is the significance level from the Mahalanobis distance test.

The exploratory analysis included a sensitivity analysis conducted to screen for relevant potential confounding variables in harmony with the procedure adopted elsewhere [25, 28]. In this analysis, univariate regression models were used to estimate crude coefficients (i.e. standardized and unstandardized coefficients and their 95% confidence intervals) indicating the influence of the covariates and NPAC on each of the healthcare performance indicators. Covariates with p>0.25 were removed from the analysis and the remaining ones kept for the second level of the sensitivity analysis. At this stage, only NHIS status was removed. At the second level, multiple linear regression models were fitted to estimate coefficients (including their 95% confidence intervals) representing the influences of NPAC and each of the covariates on each of the performance indicators. Any covariate that led to a 10% change (decrease or increase) in the coefficients of the performance indicators from the first level was kept and incorporated into the structural model as a covariate. At this stage, gender and educational level were retained.

In the second phase, a structural model was fitted to assess both direct and indirect associations, providing a basis for the mediation analysis. Fig 1 shows the structural model with six relationships (i.e., $H_1$ to $H_6$) tested. The figure's footnote provides relevant details. The mediation influences of care quality and patient satisfaction on the association between NPAC and patient loyalty were conducted following a previous study [21]. Statistical significance of associations was detected at p<0.05.

## Results

Table 1 summarizes patient characteristics. Of the 605 patients who responded, 60% (n = 363) of them were female whereas 40% (n = 242) were male. Moreover, 27% (n = 165) of the

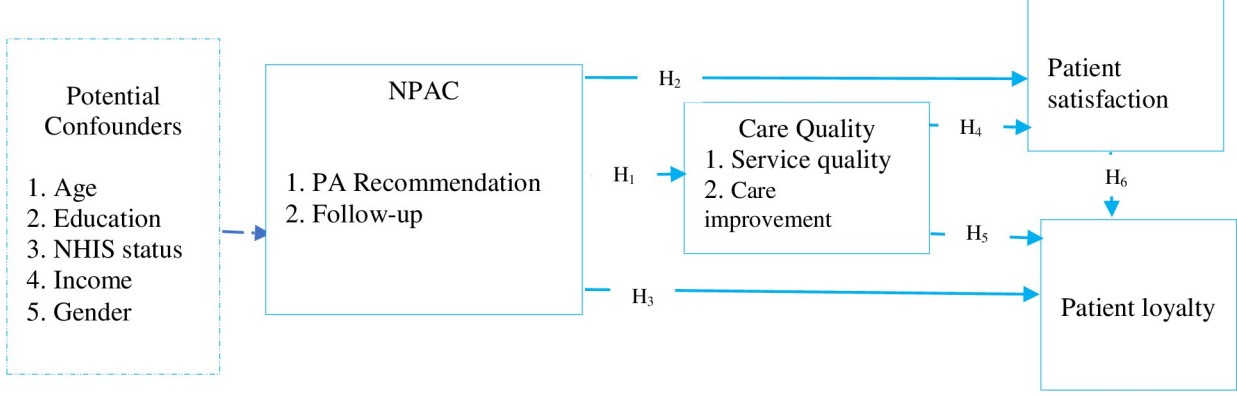

**Fig 1. A framework of the relationships between NPAC, care quality, patient satisfaction, and loyalty. Note**: NPAC–nurses' physical activity counselling, PA–physical activity; NHIS–National Health Insurance Scheme; broken arrow from the potential confounders represents confounding; $H_1$ –the association between NPAC and care quality; $H_2$ –the association between NPAC and patient satisfaction; $H_3$ –the association between NPAC and patient loyalty; $H_4$ –the association between care quality and patient satisfaction; $H_5$ –the association between care quality and patient loyalty; $H_6$ –the association between patient satisfaction and loyalty.

patients had basic education; 48% (n = 289) of them had secondary education; and 25% (n = 151) had tertiary education. About 71% (n = 430) of patients were NHIS subscribers while 29% (n = 175) were not. The average age of patients was about 35 years (Mean = 35.21; SD = 3.21) while average income was about 900 (Mean = 900.32; SD = 12.09). In Table 2, the average NPAC is about 13 (Mean = 13.33; SD = 2.36) whereas the average patient satisfaction is about 6 (Mean = 6.3; SD = 3.32). The average patient loyalty and care quality is about 10 (Mean = 9.82; SD = 2.44) and 93 (Mean = 92.73; SD = 18.53) respectively.

In Table 3, NPAC is positively correlated to care quality ($r$ = 0.337; $p$ = 0.000; two-tailed) and patient loyalty ($r$ = 0.217; $p$ = 0.000; two-tailed) but not patient satisfaction ($r$ = 0.049; $p$ > 0.05; two-tailed). This result suggests that care quality and patient loyalty increase as NPAC increases. Care quality is also positively correlated with patient satisfaction and loyalty, while patient satisfaction and loyalty are significantly correlated.

In Table 4, NPAC has a positive direct association with care quality ($\beta$ = 0.337; CR = 8.65; $p$ = 0.000) but not with patient satisfaction ($\beta$ = 0.01; CR = 0.22; $p$ > 0.05) and patient loyalty ($\beta$ = 0.046; CR = 1.21; $p$ > 0.05). This result confirms that the quality of care increases as NPAC increases in healthcare. NPAC has an indirect positive association with patient satisfaction through care quality ($\beta$ = 0.035, p <0.05). Care quality also has an indirect association with patient loyalty through patient satisfaction ($\beta$ = 0.019, p <0.05). Last but not least, NPAC has an indirect positive association with patient loyalty through care quality ($\beta$ = 0.16, p

**Table 1. Psychometric indicators of the NPAC scale.**

| Construct/scale | Factors | CA | AVE | MSV | ASV |
|---|---|---|---|---|---|
| NPAC | PA recommendation | 0.753 | 0.570 | 0.153 | 0.095 |
| | Follow-up | 0.822 | 0.623 | 0.153 | 0.104 |
| | Whole scale | 0.746 | --- | --- | --- |
| HEALTHQUAL | Service quality | 0.889 | --- | --- | --- |
| | Care improvement | 0.926 | --- | --- | --- |

† Model fit statistics for the measurement model: $\chi2$ = 2.328; p = 0.120; GFI = 0.977; TLI = 0.985; RMSEA = 0.038. ꟼ—Value not applicable. NPAC–Nurses' physical activity counselling; CA–Cronbach's alpha; AVE–average variance extracted; MSV–maximum shared variance; ASV–average shared variance

**Table 2. Summary statistics on patient characteristics (*n* = 605).**

| Variable | Level | Frequency[a]/Mean[b] | Percent[a]/SD[b] |
|---|---|---:|---:|
| Gender | Male | 242 | 40% |
| | Female | 363 | 60% |
| | Total | 605 | 100% |
| Educational level | Basic | 165 | 27% |
| | Secondary | 289 | 48% |
| | Tertiary | 151 | 25% |
| | Total | 605 | 100% |
| NHIS subscription | Subscriber | 430 | 71% |
| | Non-subscriber | 175 | 29% |
| | Total | 605 | 100% |
| Age (years) | --- | 35.21 | 3.21 |
| Income (₵) | --- | 900.32 | 12.09 |

⁻Not applicable.

[a.]for categorical variables

[b.]for continuous variables. SD–standard deviation

<0.05) and patient satisfaction (β = 0.15, p <0.05). Under Table 3, model fit indices meet the following recommended criteria: $\chi^2 \geq 3$; $p \geq 0.05$; GFI $\geq 0.95$; TLI $\geq 0.9$; RMSEA $\leq 0.08$ [22, 27]. The fit of the structural model through which the relationships were tested was therefore satisfactory.

## Discussion

This study examined the associations between NPAC, care quality, patient satisfaction, and patient loyalty. It confirmed a positive association between NPAC and care quality, suggesting that higher patients' perceived care quality was associated with higher NPAC. This result is consistent with the Social Exchange Theory (SET) originally developed by Homans [29]. The SET argues that people develop relationships based on a cost-benefit analysis in which they compare their benefit in the relationship to their cost. If the benefit exceeds the cost the relationship is deemed rewarding and valuable by the individual. Rational individuals would, therefore, stick to a rewarding relationship as long as possible. Research has found that patients consider PA a healthy behavior that benefits individual health [7, 17], and because health is a basic need that the individual would want to maintain over time health care including PA counselling would be valued by patients. That is, PA guidance from nurses would support the

**Table 3. Descriptive statistics and bivariate correlations of relevant variables (*n* = 605).**

| Variable | Mean | SD | # | 1 | 2 | 3 | 4 | 5 | 6 |
|---|---:|---:|---:|---:|---:|---:|---:|---:|---:|
| NPAC | 13.33 | 2.36 | 1 | 1 | 0.049 | .217** | .337** | -.108** | .174** |
| Patient satisfaction | 6.30 | 3.32 | 2 | | 1 | .256** | .142** | -.203** | -0.037 |
| Patient loyalty | 9.82 | 2.44 | 3 | | | 1 | .492** | -.245** | .199** |
| Care quality | 92.73 | 18.53 | 4 | | | | 1 | -.291** | .206** |
| Gender (female) | 0.59 | 0.49 | 5 | | | | | 1 | -.160** |
| Education | 1.94 | 0.71 | 6 | | | | | | 1 |

**p<0.001

*p<0.05 SD–standard deviation; NPAC–nurses' physical activity counselling

**Table 4. The relationships between NPAC, care quality, patient satisfaction, and patient loyalty (*n* = 605).**

| Dependent variable | Path | Independent variable | Coefficients | | SE of B | CR | Coefficients | |
|---|---|---|---|---|---|---|---|---|
| | | | B | β | | | Indirect β | Total β |
| *Main coefficients* | | | | | | | | |
| Care quality | <--- | NPAC | 2.644 | 0.337 | 0.306 | 8.654** | --- | 0.337** |
| Patient satisfaction | <--- | NPAC | 0.013 | 0.009 | 0.060 | 0.217 | 0.035* | 0.044** |
| Patient loyalty | <--- | NPAC | 0.047 | 0.046 | 0.038 | 1.214 | [0.16*][a][0.15*][b] | 0.195** |
| Patient satisfaction | <--- | Care quality | 0.018 | 0.103 | 0.008 | 2.299* | --- | 0.103** |
| Patient loyalty | <--- | Care quality | 0.054 | 0.417 | 0.005 | 10.571** | 0.019* | 0.436** |
| Patient loyalty | <--- | Patient satisfaction | 0.137 | 0.188 | 0.026 | 5.198** | --- | 0.188** |
| *Covariate coefficients* | | | | | | | | |
| NPAC | <--- | Education | 0.536 | 0.161 | 0.137 | 3.904** | | |
| NPAC | <--- | Gender (female) | -0.395 | -0.082 | 0.197 | -2.004* | | |
| Patient satisfaction | <--- | Education | -0.42 | -0.09 | 0.195 | -2.16* | | |
| Patient satisfaction | <--- | Gender (female) | -1.254 | -0.187 | 0.284 | -4.409** | | |
| Patient loyalty | <--- | Gender (female) | -0.333 | -0.068 | 0.184 | -1.811** | | |
| Patient loyalty | <--- | Education | 0.355 | 0.105 | 0.124 | 2.855** | | |

[a]Indirect influence of NPAC on patient loyalty through care quality

[b]Indirect influence of NPAC on patient loyalty through patient satisfaction;—Value not applicable.

**p<0.001

*p<0.05. CR–critical ratio; B–unstandardized effect; β–standardized effect; SE–standard error. **Model fit indices**: Chi-square ($\chi^2$) = 1.321; p = 0.211; goodness-of-fit index (GFI) = 0.981; Tucker-Lewis index (TLI) = 0.933; root mean square error of approximation (RMSEA) = 0.041.

health of patients and would, as a result, be highly rated by patients. As such, patients' care quality rating would increase as PA counselling in a clinical setting increases. Supporting this reasoning and the above result are studies [17, 30] that have found that patients value PA counselling and expect PA counselling from their frontline caregivers.

According to the study, NPAC has no significant direct association with patient satisfaction and loyalty. This result formed the basis of the full mediation influence of care quality and patient satisfaction on the association between NPAC and patient loyalty. This full mediation suggests that NPAC has a positive association with patient loyalty owing to care quality and patient satisfaction. In other words, patient loyalty results from NPAC only when NPAC improves perceived care quality and satisfaction. If so, the influence of NPAC on care quality is the ideal foundation for the incremental influence of NPAC on patient loyalty. Hence, the influence of NPAC on patient loyalty is not independent of care quality and patient satisfaction in the sense that NPAC must predict care quality to result in patient satisfaction and loyalty. This result supports the foregoing adaption of the SET which implies that care quality is a reward that would encourage patients to continue using a particular health care facility. It is also corroborated by studies [13, 31] that have revealed that PA counselling in health care impels patients to revisit their frontline caregivers in the hospital to report progress and challenges faced in a new routine of PA.

Failure of PA to predict patient satisfaction and loyalty directly may be owing to the fact that nurses are not suited for PA counselling [13, 18, 32], especially in a developing country where healthcare professionals are not trained to provide exercise counselling [18, 31]. For lacking technical skills relevant to PA counselling, nurses may fail to give the right information to patients, which could lead to dissatisfaction or failure of PA counselling to directly improve patient satisfaction and encourage patients to return to the hospital. Moreover, PA counselling can result in dissatisfaction and/or the decision of patients to stop using a particular health

facility if it results in harmful PA [13, 26]. Harmful PA in this context has been defined as any form of physical activity resulting in musculoskeletal injuries and dislocations [33]. It is rational for a patient to quit a new routine of PA and withdraw from any facility that recommended this lifestyle if it results in such injuries. With this in mind, the effort of some countries to equip nurses with PA counselling skills is laudable and would have to be emulated by developing countries.

We acknowledge that this study has a number of limitations. Firstly. It was based on a sample drawn from a single health facility and employed a correlational technique that is not robust enough against confounding variables. The replication of this study in other populations using experimental designs such as randomized controlled interventional trials is therefore imperative. With experimental designs, future researchers can demonstrate the causal effect of NPAC on health care performance indicators. Our utilization of some selection criteria to recruit participants made our sampling procedure non-probabilistic, which means that our sample is not necessarily representative of the general population. Our minimum sample size calculation may, nevertheless, compensate for this limitation. Despite the above limitations, this study is the first to examine the influence of NPAC on health care performance outcomes and therefore demonstrates whether PA or exercise counselling in health care is well aligned with the mission of health care facilities, which is to deliver quality and satisfactory care. Given the above findings, it is understandable that PA counselling in health care is a step toward addressing patients' needs and is therefore in line with the mandate of hospitals. In harmony with increasing advocacy for clinical PA counselling [7, 30, 32], therefore, the rolling out of a policy emphasizing PA counselling in health care could be a step in the right direction.

## Conclusion

Higher NPAC is positively associated with care quality, patient satisfaction, and loyalty, which suggests that the incorporation of PA counselling in clinical nursing can meet patients' quality expectations and encourage long-term utilization of nursing care. The adoption of PA counselling in clinical nursing can, thus, be consistent with the core mandate of health facilities, which is to deliver sustainable quality care. This being so, health facilities in Ghana and other developing countries not yet practicing PA counselling can adopt and rollout a policy of PA counselling in clinical nursing. This policy should emphasize a need for nurses to be regularly trained to provide PA counselling. More so, formal education and training of nurses must be designed to impart expertise relevant to PA counselling. Future studies replicating this study in other settings can enhance our evidence and opportunities for adopting the foregoing policy.

## Supporting information

**S1 Appendix. Items and dimensions of NPAC.**
(DOC)

**S2 Appendix. Items and dimensions of care quality indicators.**
(DOC)

**S1 Questionnaire. Inclusivity in global research.**
(DOC)

**S1 Raw data.**
(SAV)

## Author Contributions

**Conceptualization:** Nestor Asiamah.

**Data curation:** Nestor Asiamah.

**Formal analysis:** Nestor Asiamah.

**Funding acquisition:** Emmanuel Opoku.

**Investigation:** Nestor Asiamah.

**Supervision:** Nestor Asiamah, Kyriakos Kouveliotis.

**Validation:** Emmanuel Opoku, Kyriakos Kouveliotis.

**Visualization:** Kyriakos Kouveliotis.

**Writing – original draft:** Nestor Asiamah.

**Writing – review & editing:** Nestor Asiamah, Emmanuel Opoku, Kyriakos Kouveliotis.

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
