## [Decision Letter · Decision Letter 0]

21 Apr 2022

PONE-D-22-06714The association between nurses’ physical activity counselling and patients’ perceptions of care quality in a primary care facility in GhanaPLOS ONE

Dear Dr. Nestor Asiamah,

Thank you for submitting your manuscript to PLOS ONE. After careful consideration, we feel that it has merit but does not fully meet PLOS ONE’s publication criteria as it currently stands. Therefore, we invite you to submit a revised version of the manuscript that addresses the points raised during the review process.

We look forward to receiving your revised manuscript.

Kind regards,

María del Carmen Valls Martínez, Ph.D.

Academic Editor

PLOS ONE

Journal Requirements:

 [NO]. 

Additional Editor Comments:On page 10 of the manuscript, the authors talk about "customer satisfaction". It would be more appropriate to say "patient satisfaction". It is true that the customer in the healthcare system is the patient. However, it is more accurate to speak directly of the patient.The authors sometimes use "counselling" and sometimes "counselling". Please decide on one form and always use it.It would be useful to make a graph showing the relationships between the variables, so that the reader can better follow the text. You can see graphs of this type in many articles. You can see what I mean in the following article (it is not necessary to use it as a bibliographical reference): https://doi.org/10.3390/ijerph18052304

Reviewers' comments:

Reviewer's Responses to Questions

**Comments to the Author**

1. Is the manuscript technically sound, and do the data support the conclusions?

Reviewer #1: Yes

Reviewer #2: Yes

2. Has the statistical analysis been performed appropriately and rigorously? 

Reviewer #1: Yes

Reviewer #2: Yes

3. Have the authors made all data underlying the findings in their manuscript fully available?

Reviewer #1: Yes

Reviewer #2: Yes

4. Is the manuscript presented in an intelligible fashion and written in standard English?

Reviewer #1: Yes

Reviewer #2: Yes

5. Review Comments to the Author

Reviewer #1: I am delighted to review this paper. The study is important for the patients in fast recovery, especially post-operative patients related to bone fracture, nerve injury, delivery etc, by decreasing the burden of disease and health care costs as a result increasing productivity and development.

Despite the good nature of the paper, here are my concerns below.

1. Study population: your study population was patients in the OPD, is that applicable to practice PA counseling for all patients came OPD? I think PA counseling for patients having specific diseases like hypertension, DM etc. Authors study population should be clear

2. Method:” Covariates with p>0.25 were removed and those with p≤0.25 were kept for the second level of the sensitivity analysis” no need to write in this way it is better to write Covariates with p≤0.25 were included for the multivariate analysis (second analysis)

3. Results: the description of the table should be written above the table and cited in the manuscript. See journal criteria for table presentation, in the mean while in the result part write only the main findings, rather presenting all the findings from the table.

4. Conclusion: The conclusion in the abstract and in the manuscript is similar. This section should not repeat what has been mentioned in the abstract; rather, the research and policy importance obtained from the finding should be pointed out.

Furthermore, significant text overlap in method and discussion part seen in your manuscript E.g. https://www.researchgate.net/publication/303948025_HEALTHQUAL_a_multi-item_scale_for_assessing_healthcare_service_quality, www.researchgate.net › profile › Nestor-AsiamahThe Influence of Physicians’ Physical Activity Prescription, https://www.researchgate.net/publication/342601951_The_Influence_of_Physicians%27_Physical_Activity_Prescription_on_Indicators_of_Health_Service_Quality/ and, www.ncbi.nlm.nih.gov › pmc › articlesA Randomised Controlled Trial of Triple Antiplatelet Therapy etc. I would recommend that the authors carefully re-check these.

Reviewer #2: I am an academic teacher and for years has been preparing nursing staff to work with patients.I am convinced that the nursing staff is prepared for the programphysical activity counselling.Your article confirms this belief.You rightly emphasize the role of p[atient loyality. The research methodology is correct and the conclusions are well documented.I believe that the article is important and necessary, I have no reservations about publishing it.

6. PLOS authors have the option to publish the peer review history of their article (what does this mean?). If published, this will include your full peer review and any attached files.

Reviewer #1: No

Reviewer #2: No

---

## [Author Response · Author response to Decision Letter 0]

12 May 2022

Response to the Editors and Reviewers 

We are indeed grateful to the editors and reviewers for their expert comments. Despite disruptions from COVID-19, you were able to provide timely feedback. Above all, the comments are extremely helpful; we have incorporated all of them into our revision. Our responses to the comments are highlighted for your easy tracking. 

Additional Editor Comments: 

Comment: On page 10 of the manuscript, the authors talk about "customer satisfaction". It would be more appropriate to say "patient satisfaction". It is true that the customer in the healthcare system is the patient. However, it is more accurate to speak directly of the patient.

Our response: We agree with you completely. We have changed that wording. 

Comment: The authors sometimes use "counselling" and sometimes "counselling". Please decide on one form and always use it.

Our response: This is true. We have used ‘counselling’ throughout the manuscript. 

Comment: It would be useful to make a graph showing the relationships between the variables, so that the reader can better follow the text. You can see graphs of this type in many articles. You can see what I mean in the following article (it is not necessary to use it as a bibliographical reference): https://doi.org/10.3390/ijerph18052304

Our response: We are very grateful for this. We thought as much. Figure 1 has been added as the structural model with all the relationships tested. By providing the structural (hypothetical) model, we are able to depict the role of the potential confounding variables, including those removed in the sensitivity analysis, in the study. Thanks once again for this. 

Reviewer #1

Comment: I am delighted to review this paper. The study is important for the patients in fast recovery, especially post-operative patients related to bone fracture, nerve injury, delivery etc, by decreasing the burden of disease and health care costs as a result increasing productivity and development.

Despite the good nature of the paper, here are my concerns below.

Our response: Your comments are highly relevant. Thanks 

Comment: 1. Study population: your study population was patients in the OPD, is that applicable to practice PA counseling for all patients came OPD? I think PA counselling for patients having specific diseases like hypertension, DM etc. Authors study population should be clear

Our response: Thanks for this. We have provided details in the manuscript under methods (selection). Only outpatients met the inclusion criteria as in-patients were isolated and did not have the opportunity to practice PA counselling. Other reasons can be found in the manuscript. 

Comment: 2. Method:” Covariates with p>0.25 were removed and those with p≤0.25 were kept for the second level of the sensitivity analysis” no need to write in this way it is better to write Covariates with p≤0.25 were included for the multivariate analysis (second analysis)

Our response: Yes, this is logical indeed. We’ve revised that part. 

Comment: 3. Results: the description of the table should be written above the table and cited in the manuscript. See journal criteria for table presentation, in the mean while in the result part write only the main findings, rather presenting all the findings from the table.

Our response: Thanks for drawing our attention to this. We’ve rearranged the tables. 

Comment: 4. Conclusion: The conclusion in the abstract and in the manuscript is similar. This section should not repeat what has been mentioned in the abstract; rather, the research and policy importance obtained from the finding should be pointed out.

Our response: We agree with you. We’ve rewritten the conclusion section. 

Comment: Furthermore, significant text overlap in method and discussion part seen in your manuscript E.g. https://www.researchgate.net/publication/303948025_HEALTHQUAL_a_multi-item_scale_for_assessing_healthcare_service_quality, www.researchgate.net › profile › Nestor-AsiamahThe Influence of Physicians’ Physical Activity Prescription, https://www.researchgate.net/publication/342601951_The_Influence_of_Physicians%27_Physical_Activity_Prescription_on_Indicators_of_Health_Service_Quality/ and http://www.ncbi.nlm.nih.gov › pmc › articlesA Randomised Controlled Trial of Triple Antiplatelet Therapy etc. I would recommend that the authors carefully re-check these.

Our response: This is an important observation. We’ve revised parts of the methodology and discussion to reduce the similarity. 

Reviewer #2: 

Comment: I am an academic teacher and for years has been preparing nursing staff to work with patients.I am convinced that the nursing staff is prepared for the programphysical activity counselling.Your article confirms this belief.You rightly emphasize the role of p[atient loyality. The research methodology is correct and the conclusions are well documented.I believe that the article is important and necessary, I have no reservations about publishing it.

Our response: We are indeed grateful for your time and priceless contribution to this manuscript.

---

## [Editor Report · Decision Letter 1]

7 Jun 2022

The association between nurses’ physical activity counselling and patients’ perceptions of care quality in a primary care facility in Ghana

PONE-D-22-06714R1

Dear Dr. Nestor Asiamah,

We’re pleased to inform you that your manuscript has been judged scientifically suitable for publication and will be formally accepted for publication once it meets all outstanding technical requirements.

Kind regards,

María del Carmen Valls Martínez, Ph.D.

Academic Editor

PLOS ONE
---

## [Editor Report · Acceptance letter]

12 Jul 2022

PONE-D-22-06714R1 

The association between nurses’ physical activity counselling and patients’ perceptions of care quality in a primary care facility in Ghana 

Dear Dr. Asiamah:

I'm pleased to inform you that your manuscript has been deemed suitable for publication in PLOS ONE. Congratulations! Your manuscript is now with our production department. 

Kind regards, 

on behalf of

Dr. María del Carmen Valls Martínez 

Academic Editor

PLOS ONE